# Quality of Life and Health: Influence of Preparation for Retirement Behaviors through the Serial Mediation of Losses and Gains

**DOI:** 10.3390/ijerph16091539

**Published:** 2019-04-30

**Authors:** María Dolores Hurtado, Gabriela Topa

**Affiliations:** 1Campus Las Lagunillas, s/n. Edificio C-5. Despacho No. 128 C.P., Jaen University, 23071 Jaén, Spain; 2Department of Social and Organizational Psychology, The National Distance Education University (UNED), 28040 Madrid, Spain

**Keywords:** retirement, loss of resources, gains of resources, adjustment to retirement, quality of life, health

## Abstract

The dynamic theory of resources is a recent approach that provides a theoretical framework for understanding, forecasting, and examining the relationships between people’s resources and their adaptation to retirement. This article focuses on the transition to retirement in order to better understand how retirees’ perceptions of their gains and losses when they approach retirement significantly explain their well-being after retirement. Moreover, we explore the relationship between people’s preparation behaviors before retirement (T1) and their quality of life and health after retirement (T3), taking into consideration the mediating role of perceived gains and losses in retirement (T2). This study was carried out with a sample of Spanish workers (*N* = 244) who were employed at T1 and had retired at T2 and T3. The results support the assertion that losses explain well-being better than gains. In addition, some specific losses showed a greater explanatory power for quality of life and health than others. The implications are discussed with a view to understanding retirement and the design of interventions.

## 1. Introduction

As the population of developed countries ages, empirical studies on older people’s quality of life and health are increasing. However, many of them evaluate the adjustment to retirees’ new situation globally, or they do so without accounting for the mediators that intervene in these processes [1]. “Adjustment to the new status of retirees is reached when people are no longer anxious about the transition to retirement but feel comfortable and at ease with the changes that have occurred in their lives [2]”. Because of such changes, adaptation to retirement should be understood more as a dynamic process of adjustment between the person and the environment than as a final outcome [3].

In recent years, several experts have suggested applying the dynamic resource-based perspective to thoroughly explore the nature of this adjustment and to propose hypotheses that are verifiable through empirical research [4]. The key premise of this perspective is that the final adjustment people can achieve is the result of their personal access to resources. In this context, resources are considered to be all those material or immaterial elements that help people to achieve their goals. In this sense, in order to understand the adjustment process, researchers should explore not only the outcomes but also the fluctuations of resources that can influence these outcomes.

Empirical studies reveal great heterogeneity between retirement experiences, which have been synthesized in various meta-analyses [5]. Despite this heterogeneity, almost all authors agree that preparation for retirement is a variable of great relevance for the final adaptation [6]. Thus, this study tests a model in which retirement preparation behaviors occur when people are still working (T1), along with the losses and gains of resources when people are already retired (T2), and will explain subsequent adjustment (T3), directly through well-being indicators such as quality of life and health. Whereas many retirement studies have used cross-sectional designs, in the present work, an effort is made to examine the variables at different time periods, applying the resource-based dynamic perspective.

### 1.1. Antecedents of Retirement Adjustment

In the transition from work to retirement, people experience great changes in the surrounding environment, in themselves, and in their adjustment to the environment, and all these transformations can reduce their psychological comfort about retirement [7]. People’s effort to re-establish congruence with the environment in this new phase leads to diverse outcomes, such as the increase in available resources, quality of life, and health, which are considered as indicators of adaptation to retirement.

Though a wide range of antecedents could influence retiree adjustment outcomes, the resource-based perspective emphasizes that adaptation to transitions depends on the relationship between the gains and losses of resources experienced during the process. In more detail, it can be said that the balance of resources, including both the real and the perceived balance, should be considered to better understand retiree adaptation to the transition.

Firstly, if people perceive that their resources have decreased in a vital facet, then they will be forced to devote additional resources to compensate for the loss, even if such a loss is not real. The amount of resources is expected to decrease after retirement. As retirees no longer have regular income after retirement [8], their financial security is reduced [9]. They also lose social identity and self-worth derived from work, and they reduce their frequency of contact with former coworkers [10,11]. Moreover, retirees face a number of challenges, such as age-related changes in physical and cognitive health [4]. On the contrary, if people have available resources that had been devoted to their job during their working life and now are their own, when they retire, they will be able to invest these resources for other purposes. This change could be aimed at compensating for possible losses associated with retirement—managing one’s investments or redistributing these inactive resources by initiating volunteerism or caring for grandchildren, for example.

Secondly, resource fluctuations can increase the perceptions of retirement-related losses at a particular time. However, the perception of these losses can be offset in the future. In this sense, the perceptions of real losses per se can explain retirees’ overall rating of adjustment, as the adjustment process is dynamic and the outcomes achieved cannot be considered fixed and definitive in time. These fluctuations are therefore a key element of this perspective, considering the theoretical framework of the resource-based dynamic model [12,13].

The literature has shown a notable heterogeneity of outcomes related to the antecedents of retirement adjustment. A comprehensive review of the topic [1], applying the dynamic resource-based perspective, identified the main determinants of adjustment. These can be summarized as:(1)Variables related to the transition to retirement;(2)psychological predispositions;(3)changes in resources.

Psychological predispositions have been explored in other recent works which have focused on specific self-efficacy for retirement [14] or on optimism [15], so the present study will not explore these variables.

### 1.2. Individual Retirement-Related Features

They have generally been included as control variables, because age, gender, job seniority, and family characteristics may show moderate correlations with retirement adjustment at the individual level. Age is directly related to the social transition called retirement [16]. Age can also be considered a variable that is associated with other variables such as social norms regarding retirement or stereotypes prevailing in a given context about older people [17,18,19], which are also distal antecedents of adaptation.

Seniority in the post or workplace tenure has also been considered a distal antecedent of adjustment for three reasons. Firstly, because seniority is a reflection of financial resources, since greater seniority implies more contributions to the future pension [20]. Secondly, due to the fact that workplace tenure is directly related to knowledge about the job, which is a cognitive resource that could serve to control the environment [12]. Thirdly, seniority in the post should be considered a variable that masks the influence of attitudinal influences in retirement adjustment such as job attachment or engagement, which have been shown to be an antecedent to retirement adjustment [21].

On another hand, gender, which has been repeatedly explored in relation to retirement, shows complex relationships with both resources and adjustment. While gender-based differences in financial preparation seem to favor males, the advantages of the adjustment achieved after retirement seem to favor women. Likewise, as has been repeatedly indicated in the literature, family burdens—either in the form of a greater number of economically dependent persons or of relatives who require physical care—may influence retirement adjustment. However, the discussion of these issues exceeds the scope of this work. Based on this literature, in the present study, age, seniority in the post, gender, and the number of economically dependent persons will be used as control variables in order to statistically isolate their influence on retirees’ adjustment.

### 1.3. Retirement Preparation Behaviors

Preparation for retirement is a multidimensional construct that can be defined as “the thoughts and behaviors aimed at goals that promote good health and provide financial security, adapt lifestyles, and ensure gratifying rewards in retirement” [6]. The review of Barbosa and colleagues (2016) on retirement adjustment found that preparation activities are one of the predictors of adjustment. In particular, activities aimed at preparing the person for retirement were beneficial in 56.5% of the cases. Paradoxically, preparation was a risk factor in two cases: When discussing the subject with the family [22] and when planning one’s social life [23].

Most studies report that people usually plan financially by saving, seeking professional advice, and engaging in informal discussions and comparing themselves with others [24,25]. Some people also plan their retirement lifestyles [26] and prepare the changes in social roles (psychosocial planning) as they transit from employee to retiree [14]. Psychosocial planning, on another hand, involves thinking about new roles, talking to retirees about their experiences, and distancing oneself from the worker role [6]; however, it also implies managing new social relations and looking for paid or unpaid activities with which to occupy one’s time and contribute to one’s satisfaction. Though not exclusively linked to retirement, some people also participate in positive health behaviors (increasing physical activity, for example) to protect their long-term health [22,27]. Regardless of the use of global measures, specific single-aspect measures, or multiple measures to cover all the facets of preparation, it is agreed that retirement preparation appears to be a predictor of medium- and long-term results.

Based on the literature reviewed to date, in the present study, these hypotheses are proposed:

**Hypothesis** **1** **(H1).**
*Retirement preparation behaviors (T1) are expected to positively predict retirees’ quality of life of (T3).*


**Hypothesis** **2** **(H2).**
*Retirement preparation behaviors (T1) are expected to positively predict retirees’ health (T3).*


### 1.4. Gains and Losses of Resources

The *Conservation of Resources* (COR) theory [28], which is a general approach to stress, has been applied to a wide range of vital moments and is also the remote antecedent of the resource-based dynamic perspective to understand retirement. The basic principle of COR is that people make a great effort to maintain and accumulate resources, that losses have harmful effects on them, and that gains are protective. When faced with stressful events, people try to minimize the loss of resources, as the actual loss itself is enough to produce stress. Losses are important in two ways. Firstly, resources have an instrumental value for people, and secondly, they have a symbolic value, as they help people define themselves and others. The psychological stress associated with the transition from work to retirement is explained as a reaction to the environment in the face of a loss of resources.

When people are not facing these events, they are dedicated to accumulating additional resources [28]. The gains in resources also play a role in well-being; unfortunately, their influence is much lower than that of losses. The COR affirms and empirical studies support this premise,—that the loss of resources is much more influential and more powerful than gains [29,30,31]. This is due to a number of reasons. Firstly, it is more difficult to prevent loss than to obtain gains, and when loss occurs, the decrease in resources is greater than any associated gain. Secondly, loss is not only disproportionate in terms of degree, but also of speed. The economic crisis seems like an eloquent example of how the seemingly solid structures need very little time to fall [30]. Finally, loss, due to its power to seriously threaten survival, has a much more important information value than gains from an adaptive point of view.

Gains are important when they prevent future losses and, to a lesser extent, when they provide well-being. Gain strategies, such as accumulating food or investing in the social group, are significant because, firstly, they imply that any loss that occurs will not be translated immediately and inevitably into a critical state of resources that could place people in situations of survival risk. Secondly, gains produce comfort. This would explain why, though shelter and defense or physical protection are basic resources, people try to buy luxurious and comfortable homes. These luxuries are secondary because we can survive without them, but we nonetheless acquire them because they increase our comfort [31]. Despite this enriching value of gains, the COR postulates that the predictive power of losses on people’s well-being will always be higher than that of gains.

What seems clear from the focus of the COR is that gains and loss are not independent of each other; they are related. As Hobfoll has repeatedly stated, resources “travel in caravans,” and therefore, the influence of a fluctuation, positive or negative, would unleash a spiral in which other resources could be compromised. This theoretical postulate supports the claim that losses and gains will influence the relationship between preparation and adjustment, as well as our consequent choice of the serial mediation model.

It is also necessary to consider that, as they grow older, people tend to be more motivated to conserve their resources with a view to future losses rather than to gain new resources, which allows them to preserve their functioning, [32,33]. According to this approach [34], it was found that young adults move primarily by gains, whereas older adults do so by goals related to the maintenance of their functions or to the avoidance of losses. Thus, Freund and Baltes [35] showed that, at very advanced ages, although investments in resources to compensate for losses continue to positively influence adequate functioning, compensation efforts decline with age [36]. Taken together, these findings underline the importance of avoiding losses as people age [33].

There are discrepant positions about the variability of resources and their evaluation. Resources, understood from the original formulation of the COR, may be: (1) Objects (e.g., adequate income, savings), (2) conditions (e.g., relations with one’s children, a good marriage), (3) personal characteristics (self-esteem, social competence) and (4) energies (e.g., people to learn from, affection for others, loyalty, specific aids). However, the taxonomy of resources is one of the most debated aspects of this theory. Specifically, “the dynamic resource-based perspective established that retirement adjustment depends on the physical, economic, social, and psychological (cognitive, emotional, and motivational) resources and on changes in these resources during transition to retirement [1]”. With regard to how to proceed with the assessment of resources, there is ongoing debate about the adequacy of measuring gains and losses through standardized instruments such as the COR-e [37,38], which contains an extensive list of resources, some of which may be irrelevant in the specific situation that is being assessed; or, one could select resource subsets ad hoc, which are presumed to be relevant only to the specific context. In the present study, the physical, economic, social, and psychological resources of people approaching retirement [2] are evaluated by dimensions using the standardized instruments provided by the theory of Hobfoll. Numerous studies have been carried out, applying COR to very varied fields. Hobfoll and Wells wrote a chapter related to aging in general which contains some considerations about retirement. They pointed out, among other things, that advanced age is simply the continuation of previous life cycles, and that, although there is no doubt that there is deterioration in physical and cognitive capacities, it does not necessarily have to be a stage marked by losses [39].

In short, it seems that, on the one hand, the COR provides a broad theoretical framework to understand the process of change associated with the work-retirement transition and some of the effects associated with retirement, and it is complemented by the resource-based dynamic perspective. However, its specific application to this area through empirical studies is still incipient [14,40]. In this sense, the present work proposes to explore the influences of perceived retirement-related losses and gains on the indicators of retirees’ adjustment.

### 1.5. Adjustment to Retirement: Amplitude of the Indicators

Applying the resource-based dynamic perspective, adjustment should be considered as a process of adaptation to the changes and available resources in the new situation, whether there are losses or gains with regard to the initial situation. Van Solinge [1] has recommended that direct and indirect approaches be combined to assess adjustment. The former would consist of individual perceptions of changes in resource availability as a result of the transition to retirement. The latter, however, would assess adjustment through close or proxy measures of well-being—that is, inferred from other indicators such as health [23,41].

Consequently, firstly, this study has used perceived losses and gains of resources as a direct measurement of adjustment based on retirees’ own evaluation of the situations encountered to adapt to retirement. Secondly, based on the assumption that low levels of well-being indicate greater difficulties experienced by retirees and, in turn, reveal a poor adaptation, the present study has considered quality of life in retirement and health after retirement.

When the transition to retirement has already occurred, people must face a re-evaluation of their lives. This life re-evaluation allows people to make adjustments, changing what they consider inadequate and finding the opportunity to improve their satisfaction with the facets they rate as the most important. Therefore, many studies have included quality of life measures in retirement and others have considered them as indicators of well-being in adulthood [42,43,44,45,46] (among many others).

Several models have been used to empirically analyze the personal consequences of retirement [1]. The life cycle perspective is one that seems more efficient to account for the empirical results. It indicates that people build their life course through their choices and decisions, always limited by their available opportunities and historical and social circumstances. Empirical studies of people’s adjustment to retirement usually include quality of life [47,48,49,50] and perceived health [4,51,52] as indicators of adjustment. 

Drawing on the literature [14], in the present study, we propose that: 

**Hypothesis** **3** **(H3).**
*The relationship between retirement preparation behaviors (T1) and retirees’ quality of life (T3) will be significantly mediated by the loss of physical (h3a), cognitive (h3b), motivational (h3c), financial (h3d), social (h3e), and emotional resources (h3f), whereas gains in physical, cognitive, motivational, financial, social, and emotional resources will not be significant mediators.*


**Hypothesis** **4** **(H4).**
*The relationship between retirement preparation behaviors (T1) and retirees’ health (T3) will be significantly mediated by the loss of physical, cognitive, motivational, financial, social, and emotional resources, whereas gains in physical, cognitive, motivational, financial, social, and emotional resources will not be significant mediators.*


Figure 1 provides a visual representation of the study hypotheses.

In short, the purpose of this study is to test a model theoretically derived from the process through which perceived gains and losses during retirement are associated with post-retirement adjustment indicators. According to the resource-based model, we evaluated the effects of the initial impact of antecedents such as preparation behaviors on the perception of gains and losses, and we confirmed whether this effect would influence retirement adjustment through quality of life and health over time. Retirement, as noted above, is a vital transition involving multiple tasks. Because it affects a multiplicity of different life facets, such as finances, health, and social relations, it is reasonable to assume that people’s perception of losses or gains will not be uniform for all the dimensions. Based on this reasoning, the present study seeks to analyze not only the global predictive power of losses and gains, but also the differential impact that specific dimensions may have on retirees’ well-being. 

## 2. Materials and Methods

The participants in this study (*N* = 244) were Spanish workers who were actively employed at T1. At T2, they had been retired for the past six months, and at T3, they had been retired for 9 to 12 months. About 43% of the sample were men, and the average age of respondents was 62.08 years at T1 (*SD* = 4.88), whereas the average time spent working was 29.9 years (*SD* = 9.3) at the time of retirement. Most (43.4%) of the sample had at least basic or high school studies, and 26.6% of the participants had at least one or two dependents at home. Regarding their geographical distribution, 64.2% lived in Madrid, and 33.6% in the Valencian Community. Regarding their main activity, more than 70% had worked in the services sector, 20% in industry, and a smaller percentage in other sectors.

### 2.1. Instruments

**Retirement Preparation Behaviors (T1):** The scale of Pre-Retirement Planning Activities as used [22]. The scale consists of 19 items that describe specific behaviors that people can perform distributed in four dimensions: Financial preparation (five items), health (four items), preparation of lifestyle (three items), and psychological preparation (seven items). Participants were asked how often the described behaviors had been performed during the past year. The original response scale of the instrument was replaced by a five-point Likert-type response scale ranging from 1 (never) to 5 (very often) because, in this way, more information was collected than with the original dichotomous scale. The instrument had shown adequate reliability in previous studies α = 0.74 [23]. Following the procedure used by Yeung (2017), a global index of preparation for retirement was calculated which would allow jointly assessing all the behaviors performed. Examples of items are: “You have purchased an accident insurance policy,” “You begin to cease performing habits that are hazardous for your health,” “You discuss retirement with people about to retire.”

**Gains and losses of resources (T2):** The COR Evaluation [31] questionnaire (the version translated into Spanish) was used [40]. The complete questionnaire contains 65 items, which includes a list of resources, and it asks people to assess their real losses and gains in the period after retirement, using a five-point Likert scale ranging from 1 (no loss/gain) to 5 (total loss/gain). The list of resources is presented twice, one for the evaluation of losses and the other for gains. The resources are organized randomly and include both tangible (money, adequate clothing) and intangible elements (feeling of hope, friends’ affection). Reliability indicators for both scales were adequate. As for the dimensions of losses, the overall reliability indicator was α = 0.79. In the case of gains, we calculated reliability for the total scale, finding a value of α = 0.78. The indicators in previous research were also adequate, although somewhat higher than our findings [40].

The items in the six dimensions of resources postulated by the resource-based dynamic theory of retirement were distributed as a function of their content (physical, cognitive, motivational, financial, social, and emotional resources). Two experts in the study of retirement assessed the distribution of the items in the categories, reaching an adequate inter-judge agreement index (*r* = 0.81).

The distribution of the items in the subscales of physical, cognitive, motivational, financial, social, and emotional resources was then analyzed with the Smart PLS 3.0 [53], program to evaluate the convergent validity of the dimensions of the resources. Convergent validity was estimated with the AVE (Average Extracted Variance) of the constructs, whose values should be higher than 0.50. In the present study, the values AVE for gains of resources ranged between 0.52 and 0.65, whereas for losses, it ranged between 0.52 and 0.68. The internal consistency of the dimensions was calculated through composite reliability, which ranged between 0.77 and 0.94 for gains and between 0.76 and 0.94 for losses. The traditional reliability ratings through Cronbach alpha from ranged between α = 0.73 and α = 0.92 for gains and between α = 0.74 and α = 0.93 for losses. Examples of items were: “Think about the past and tell us whether you have experienced loss of health,” “Loss of stable work,” “Think about past and tell us whether you have had positive feelings about yourself.”

**Quality of Life (T3):** To assess this variable, the CASP-12, version 3, [54] was used, because the full version (19 items) had several problems of internal consistency and dimensionality [48]. The dimensions of control and autonomy are combined into a single dimension, and the dimensions of pleasure and realization have been shortened to include only six items [55].

To measure adults’ quality of life, the psychometric instrument identifies its main aspects, considered as properties within the aging process. The average values per country range between 33.32 in Greece and 40.48 in Switzerland. The average score for the entire sample (all countries) is 37.37. In Spain, the average value is 35.57. The mean values for Spain are lower in all the dimensions of the index than the mean of the entire sample. Examples of items are: “My age prevents me from doing the things I’d like to do,” “I hope every day,” “I am satisfied with the way my life has turned out.”

**Health (T3):** The general health questionnaire, SF 36 v2 Health Survey [56], consisting of five items, was used to evaluate this variable. Prior authorization was obtained for use of the questionnaire. The response options range from 1 (totally true) to 5 (totally false). The mean score was calculated after reversing the participants’ responses so that the highest score indicated a better overall health status. The scale is widely used in general self-reported health assessments, and its validity and reliability indicators in the Spanish version are adequate [57].

As most of the instruments were in English, several experts in retirement translated the items into the Spanish context. A subsequent translation was carried out by a native English speaker and compared to the original questionnaires. Examples of items are: “I think I get sick more easily than other people,” “I am as healthy as anyone else,” “My health is excellent”.

### 2.2. Procedure

The longitudinal study of three data collection times was designed drawing on the postulate that the distal and proximal influences in the retirement process are different [58]. The behavior of retirement preparation was assessed as a distal antecedent. The perceptions of access to resources, both of losses and gains in the different dimensions, were treated as a proximal antecedent to retirement at T2. At T3, retirement adjustment was directly assessed based on perceived quality of life and health.

In the present study, a two-stage sampling approach was used [1], in which we first selected a specific group of small and medium-sized enterprises and then invited the total population of these companies who were approaching retirement the next year to participate.

The research group sent emails to 20 organizations (SMEs and public sector organizations) to participate in a comprehensive human resources management study with staff older than 60 years. The ten organizations that responded were visited by the researchers to explain the criteria for the inclusion of the participants (current employees over 60 years old and whose retirement was scheduled for the next year, approximately). Only five organizations finally participated in the study. In the first data collection, 350 employees who were active at that time received the questionnaire, a letter explaining the purpose of the study and the data collection procedure, and an envelope to return the survey. Finally, 322 complete questionnaires were collected (92% response rate). In this questionnaire, participants who agreed to collaborate at the remaining stages of the research provided their personal email address in order to allow us to contact them in the future. Five months later, we sent a second questionnaire to the participants via email, obtaining an 85% response rate (273 surveys). Respondents who completed the second data collection also updated the information of their email addresses. Approximately six months later, the third questionnaire was sent by email and was completed by 244 retirees who participated in this phase of the survey (89% response rate).

## 3. Results

First, a descriptive analysis of the variables of the study was carried out, which showed that the losses of resources were perceived as less than the gains in all the dimensions. However, in all the dimensions of resources, losses were positively related to each other. Moreover, losses were significantly related to the behavior of retirement preparation, health, and quality of life, thus providing a preliminary support to the study hypotheses. On another hand, the gains of resources, in all their dimensions, showed practically null correlations with the behaviors of preparation as well as with health and quality of life. Table 1 shows the correlations, means, and standard deviations of the variables of the study.

### 3.1. Hypothesis Testing

To test the hypotheses of the study, we conducted several of multiple mediation analyses with bootstrapping techniques, using the macro PROCESS for SPSS developed by Andrew Hayes (2013), with Model 6 of this statistical program. In each of the analyses, the influence of age, gender, job seniority, and the number of economically dependent people was controlled for.

The direct effect of retirement preparation behaviors (T1) on retirees’ quality of life (T3) was significant and positive in all models, supporting Hypothesis 1. The total model was significant, *F*(5, 24) = 32.9, *p* < 0.01, *R** = 0.41.

The direct effect of retirement preparation behaviors (T1) on retirees’ health (T3) was significant and positive in all models, supporting Hypothesis 2. The total model was significant, *F*(5, 24) = 5.02, *p* < 0.01, *R** = 0.10, although the percentage of explained variance was low.

To test Hypothesis 3, in each of the analyses, two pairs of mediators were tested—the losses and the gains of each resource dimension—taking as criterion variable the quality of life in retirement (T3). As there were three indirect effects in all cases, we first analyzed them separately and then performed the comparisons to determine which of them was more statistically significant for the model.

### 3.2. Losses and Gains of Physical Resources (h3a)

The results show that the only significant indirect effect was through physical losses, as it did not include the value 0 in the 95% confidence intervals (preparation behaviors–physical losses– quality of life [0.02, 0.19]) (Table 2), whereas the indirect effects through the mediation of gains and of conjoint losses and gains were not significant. The results of the serial mediation account for a partial effect PM = 0.11 of the total effect. The comparison between the various effects the mediation of the loss of physical resources on quality of life was significant, both separately and when subtracting the mediations of gains (path c1) and of losses and gains conjointly (path c2). Therefore, these findings support Hypothesis 3a. In Figure 2 the non-standardized B coefficients, the significance level, and confidence interval are specified. Comparisons between the indirect effects are presented in Table 2.

### 3.3. Gains and Losses of Cognitive Resources (h3b)

The results show that the only significant indirect effect was through cognitive losses, as it did not include the value 0 in the 95 % confidence intervals (preparation behaviors–cognitive losses–quality of life [0.15, 0.33]) (Table 3), whereas the indirect effects through the mediation of gains and of losses and gains concurrently were not significant. The results of the serial mediation account for a partial effect PM = 0.37 of the total effect. The comparison between the various effects the mediation of the loss of cognitive resources on the quality of life was significant, both separately and when subtracting the mediations of the gains (path c1) and of losses and gains conjointly (path c2). Therefore, these findings support Hypothesis 3b. In Figure 3, the non-standardized B coefficients, and the significance level, and confidence interval are specified. Comparisons between the indirect effects are presented in Table 3.

### 3.4. Losses and Gains of Motivational Resources (h3c)

The results show that the only significant indirect effect was through motivational losses, as it did not include the value 0 in the 95%confidence intervals (preparation behaviors–motivational losses–quality of life [0.17, 0.34]) (Table 4), whereas the indirect effects through the mediation of gains and losses and gains concurrently were not significant. The results of the serial mediation account for a partial effect PM = 0.38 of the total effect. The comparison between the various effects the mediation of the loss of motivational resources on the quality of life was significant, both separately and when subtracting the mediations of gains (path c1) and of losses and gains conjointly (path c2). Therefore, these findings support Hypothesis 3c. In Figure 4, the non-standardized B coefficients, the significance level, and confidence interval are specified. Comparisons between the indirect effects are presented in Table 4.

### 3.5. Losses and Gains of Financial Resources (h3d)

The results show that the only significant indirect effect was through financial losses, as it did not include the value 0 in the 95% confidence intervals (preparation behaviors–financial losses–quality of life [0.12, 0.29]), whereas the indirect effects through the mediation of gains and of losses and gains concurrently were not significant. The results of the serial mediation account for a partial effect PM = 0.30 of the total effect. The comparison between the various effects the mediation of the loss of financial resources on the quality of life was significant, both separately and when subtracting the mediation of gains (path c1) and of losses and gains conjointly (Path c2). Therefore, these findings support Hypothesis 3d. In Figure 5, the non-standardized B coefficients, the significance level, and confidence interval are shown. Comparisons between the indirect effects are presented in Table 5.

### 3.6. Losses and Gains of Social Resources (h3e)

The results show that the only significant indirect effect was through social losses, as it did not include the value 0 in the 95%confidence intervals (preparation behaviors–social losses–quality of life [0.04, 0.19]) (Table 6), whereas the indirect effects through the mediation of gains and of losses and gains concurrently were not significant. The results of the serial mediation account for a partial effect PM = 0.15 of the total effect. The comparison between the various effects the mediation of the losses of social resources on the quality of life was significant, both separately and when subtracting the mediations of gains (path c1) and of losses and gains conjointly (path c2). Therefore, these findings support Hypothesis 3e. In Figure 6, the non-standardized B coefficients, the significance level, and confidence interval are shown. Comparisons between the indirect effects are presented in Table 6.

### 3.7. Losses and Gains of Emotional Resources (h3f)

The results show that the only significant indirect effect was through emotional losses, as it did not include the value 0 in the 95% confidence intervals (preparation behaviors–emotional losses–quality of life [0.12, 0.31]), whereas the indirect effects through the mediation of gains and of losses and gains concurrently were not significant. The comparison between the various effects the mediation of the losses of emotional resources on the quality of life was significant, both separately and when subtracting the mediations of gains (path c1) and of losses and gains conjointly (path c2). Therefore, these findings support Hypothesis 3f.

In Figure 7, the non-standardized B coefficients, the significance level, and confidence interval are shown. Comparisons between the indirect effects are presented in Table 7.

Finally, and to test Hypothesis 4, in each of the subsequent analyses, two pairs of mediators, were tested; the losses and the gains of each resource dimension were used as a criterion variable the retirees’ health (T3). As there were three indirect effects, in all cases, we first analyzed them separately, and then we performed the comparisons to determine which of them was more statistically significant for the model.

### 3.8. Losses and Gains of Physical Resources (h4a)

The results show that the only significant effect was an indirect effect through physical losses, as it did not include the value 0 in the 95% confidence intervals (preparation behaviors–physical losses–health), whereas the indirect effects through the mediation of gains and of losses and gains concurrently were nonsignificant. The results of the serial mediation account for a partial effect PM = 0.18 of the total effect. The comparison between the different effects of the mediation of the losses of physical resources on health was significant, both separately and when subtracting the mediations of the gains (path c1) and of the losses and gains conjointly (path c2). Therefore, these findings support Hypothesis 4a. In Figure 8 the non-standardized B coefficients, the significance level, and confidence interval are shown. Comparisons between the indirect effects are presented in Table 8.

### 3.9. Gains and Losses of Cognitive Resources (h4b)

The results show that the only significant effect was an indirect effect through cognitive losses, because it did not include the value 0 in the 95% confidence intervals (preparation behaviors–cognitive losses–health [−0.25, −0.08]) (Figure 9), whereas the indirect effects through the mediation of gains and of losses and gains concurrently were nonsignificant. The results of the serial mediation account for a partial effect PM = 0.21 of the total effect. The comparison between the different effects of the mediation of losses of physical resources on health was significant, both separately and when subtracting the mediations of the gains (path c1) and of the losses and gains conjointly (path c2). Therefore, these findings support Hypothesis 4b. In Figure 9, the non-standardized B coefficients, significance level, and confidence interval are shown. Comparisons between the indirect effects are presented in Table 9.

### 3.10. Losses and Gains of Motivational Resources (h4c)

The results show that the indirect effect through motivational losses was significant, as it did not include the value 0 in the 95% confidence intervals (preparation behaviors–motivational losses–health [−0.23, −0.08]) (Figure 10) as was the indirect effect through the losses and gains conjointly (preparation behaviors–motivational losses–motivational gains–health [−0.22, −0.009]), whereas the indirect effects through the mediation of gains were not significant. The results of the serial mediation account for a partial effect PM = 0.21 of the total effect. The comparison between the different effects of the mediation of motivational resources on health was significant, both separately and when subtracting the mediations of the gains (path c1) and of the conjoint losses and gains (path c2). Therefore, these findings support Hypothesis 4c. In Figure 10, the non-standardized B coefficients, significance level, and confidence interval are shown. Comparisons between the indirect effects are presented in Table 10.

### 3.11. Losses and Gains of Financial Resources (h4d)

The results show that the indirect effect through financial losses was significant, as it did not include the value 0 in the 95% confidence intervals (preparation behaviors–financial losses–health [−0.20, −0.07]) (Figure 11), as was the effect through losses and gains conjointly (preparation behaviors–financial losses–financial gains–health [−0.03, −0.04]), while the indirect effects through the mediation of gains were not significant. The results of the serial mediation account for a partial effect PM = 0.19 of the total effect. The comparison between the different effects of the mediation of financial resources on health was significant, both separately and when subtracting the mediations of the gains (path c1) and of the losses and gains conjointly (path c2). Therefore, these findings support Hypothesis 4d. In Figure 11, the non-standardized B coefficients, significance level, and confidence interval are shown. Comparisons between the indirect effects are presented in Table 11.

### 3.12. Losses and Gains of Social Resources (h4e)

The results show that the only significant effect was an indirect effect through social losses, because it did not include the value 0 in the 95% confidence intervals (preparation behaviors–social losses–health [−0.26, −0.12]) (Figure 12), whereas the indirect effects through the mediation of gains and of losses and gains conjointly were nonsignificant. The results of the serial mediation account for a partial effect PM = 0.18 of the total effect. The comparison between the different effects of the mediation of the losses of social resources on health was significant, both separately and when subtracting the mediations of gains (path c1) and of losses and gains conjointly (path c2). Therefore, these findings support Hypothesis 4e. In Figure 12, the non-standardized B coefficients, significance level, and confidence interval are shown. Comparisons between the indirect effects are presented in Table 12.

### 3.13. Losses and Gains of Emotional Resources (h4f)

The results show that the only significant effect was an indirect effect through emotional losses, as it did not include the value 0 in the 95% confidence intervals (preparation behaviors-emotional losses-health [−0.20, −0.07]) (Figure 13), whereas the indirect effects through the mediation of gains and of losses and gains conjointly were nonsignificant. The comparison between the different effects of the mediation of losses of emotional resources on health was significant, both separately and when subtracting the mediations of gains (path c1) and of losses and gains conjointly (path c2). Therefore, these findings support Hypothesis 4f.

In Figure 13, the non-standardized B coefficients, significance level, and confidence interval are shown. Comparisons between the indirect effects are presented in Table 13.

## 4. Discussion

In general, the objective of this study was to test the theoretical model of the process of perceived gains and losses in retirement associated with post-retirement adjustment indicators. We specifically evaluated the impact of preparation behaviors on the perception of gains and losses at T1, and we observed their impact on the quality of life and health over time (T2, T3). We considered studying this because not only has the adjustment of the changing situation of older workers been recommended [59] but also to determine what factors could help us to understand it better.

First, after analyzing the correlations between the variables of the study, we found that the losses in all the dimensions of the resources are negatively related and statistically significant to each other; that is to say that the relationship between preparation behaviors and quality of life will be mediated higher by the losses [30] than by the gains, and the same happens when we change the variable quality of life criterion by health. We find similar results in another study of the literature carried out by [60]. They showed that the gains have a significant but weak relation with the welfare of the employees, while the losses have a strong relation with the lack of welfare of the people unemployed. It seems that earnings are important when they avoid future losses and to a lesser extent when they provide welfare. Several empirical studies and some meta-analyses emphasize that access to resources, such as physical or mental health, finances, community services, and marital relationships should be considered strongly linked to the adjustment of retirement [61].

Secondly, losses and gains were analyzed using the six dimensions of resources (physical, cognitive, motivational, financial, social, and emotional) separately. In our results, we can see that, when the criterion variable is quality of life, losses affect more significantly the cognitive resources (*B =* −0.38**) and the motivational resources (*B =* −0.36**). These findings show the predictive power of resource losses to explain retirees’ quality of life. Previous studies indicate that, when the study uses quality of life, a distinction must be made between dreams and reality [62]. On another hand, quality of life is widely accepted as a useful indicator of adjustment [63]. Likewise, well-being can be considered a multidimensional construct in which we can incorporate two dimensions, hedonic and eudemonic [64], both of which proved to be predictors of health outcomes among mental health patients [65]. In recent research, cognitive resources are found to be very useful for early interpretation of stressful situations such as an abrupt or unwanted transition [66]. In addition, these involuntary situations have been associated with losses of resources (mainly of control) in empirical research [67] and in meta-analyses [68] in press. Our findings are contrary to those of [13], where the factor that included cognitive resources at T1 in their study were still significantly related to adjustment and satisfaction in retirement—although these resources were less related than at Time 2.

However, in the literature, we have found different results from those obtained for resources in the domains of cognitive abilities and motivation where there is no significant change, suggesting that retirees can maintain the resources in these two domains after retiring from their jobs. As this study only evaluates the changes in resources one year after actual retirement, long-term changes remain largely unknown [69].

Thirdly, another contribution of the study is to have included health as a criterion variable. We also saw the greater importance of losses of resources than of gains, so that the loss of social resources had a higher value (*B =* −0.19**), followed by the loss of physical resources (*B =* −0.18**). On another hand, the indirect effects show negative and significant relationships in all the mediators (physical, cognitive, motivational, financial, social, and emotional resources). Most primary studies on early retirement included health, both physical and mental, as a predictor, and concluded that poor health should be considered a powerful determinant of early retirement [70]. As shown by empirical studies [71], early retirement implies a deterioration of resources that guarantee stability, status, and privileged conditions, and the existence of valuable personal resources—knowledge, self-esteem, social contacts—that could have an impact on health [40,69]. In relation to health and loss of physical resources, we find in recent literature [72] that people out of the labor market as a result of cancer can benefit from additional support from the state, employers, and professional physicians to facilitate their return to work if they so desire. Previous studies have shown that many cancer survivors will return to work after treatment; from 40% six months to 89% 24 months after a cancer diagnosis [73,74]. Due to the loss of social resources, according to the literature, early retirees experience significant reduction in social integration, a negative experience that does not seem to be compensated through non-work activities such as volunteering, family duties, and hobbies [75]. Evidence indicates the beneficial effects of mental and social resources on post-retirement well-being. For example, perceived control and clarity of objectives (such as personal resources) are positively correlated with post-retirement well-being and adjustment [76]. In addition, the support of family and friends also predicts retirees’ well-being [77]. Consequently, the loss of valuable resources related to personal characteristics and loss of social resources such as co-workers and friends is associated with retirement. In short, in the face of retirement, it seems that the person develops a self-assessment of competences to cope with the transition, and that assessment helps minimize the perceived threat of future losses [78].

One of the advantages of this study is that many retirement studies have focused on cross-sectional designs, whereas in the present work, an effort was made to examine the variables at different temporal moments, applying the resource-based dynamic perspective. A limitation may be due to some variables, such as social, political, and economic variables, which were not assessed and which might interfere with our results. Moreover, quality of life has been assessed as a global construct, but future studies would separate it from other strictly related dimensions like endowments, mobility, life satisfaction, emotional support, and social connectedness. On another hand, retirement is an area that affects issues of family and work, and it is reasonable to assume that individual environmental factors and variables interact to facilitate or hinder access to resources in the transition to retirement, but our model does not consider such interactions. For example, in a recent article [59], it has been shown that the two specific subsets of social and personal resources interact, influencing retirement-related outcomes. Hence, we propose for future studies to carry out works and contrast hypotheses across time and compare samples in different countries, thereby establishing cross-cultural relationships. In addition, future studies should extend this research to a longer interval (for example, five years) to obtain a clear picture of changes in retirement resources over time.

The body of evidence presented [79] strongly supports the hypothesis that transitions due to lack of retirement are anticipated mainly before retirement and, for most people, they are therefore not a response to financial crises experienced after retirement, nor are they the result of poor planning or low wealth accumulation.

## 5. Practical Implications

The losses examined in our model affect people’s quality of life and health in the face of adjustment to retirement. We differentiated physical, cognitive, motivational, financial, social, and emotional resources. Therefore, we propose that, when designing seminars, courses, or workshops for retirement preparation, one should take into account the diversity of resources. However, the analysis also shows that preparation could be improved, and it would be interesting to orient interventions to maintain and increase the resources that are needed. On another hand, there is also a need to train professionals to help people who make the transition to retirement in terms of social resources and to minimize economic resource decreases [80]. We should encourage the development of motivational resources through volunteer service or other life facts associated with retirement [81]. People’s loss of motivational resources in different vital moments could also affect the relationships between the variables we have studied. Organizations should take into consideration the use of strategies to adapt jobs to older workers, thus favoring an active role [82]. Thus, older workers could take on tasks involving more counseling or mentoring skills, allowing them to pass on their knowledge to the younger workers [83]. Other strategies that could anticipate the loss of resources are to encourage creativity and helping retirees to experience the transition as an event of self-realization [84] so older workers are motivated to maintain and to stimulate their vocational interests [85]. All of this has the aim of actively pursuing leisure activities [86] that allow retirees to accumulate resources or to compensate for their losses [87].

## 6. Conclusions

People’s preparatory actions before retirement will depend on many factors. However, adaptation to retirement requires considering care comprehensively in the physical, mental, and social areas. Retirement from work implies a transition that is accompanied by losses, such as the loss of the role of worker, loss of status, and loss of social relations. The incidence of these losses on subsequent well-being will depend on personal factors, so it is essential to pay special attention to all these variables [88]. Adaptation to retirement involves knowing how to value and benefit from the greater availability of time, without having to submit to the pressures of an active working life. Though some steps have been taken in that direction, the truth is that there is still much to do. The initiative of working with people who are close to retirement is such a significant step that these people can assume the changes of aging naturally and achieve a better adaptation to this new stage, and so that society will not reject them—on the contrary, it will integrate them. In this sense, intervention in preparatory programs can facilitate adaptation to retirement so that all of these people will have an effective process of transition to this new stage.

Despite the study’s limitations and considering that most research in this area consists of cross-sectional studies, this three-moment investigation offers more depth about the psychosocial aspects related to well-being, quality of life, and health of people attempting to achieve dignified retirement adjustment. As a result, we offer more evidence of the added value of the focus of the resource-based dynamic perspective on retirees’ well-being. Our research could also serve to inform future interventions designed to focus on resource losses and to improve retirement well-being.

## Figures and Tables

**Figure 1 ijerph-16-01539-f001:**
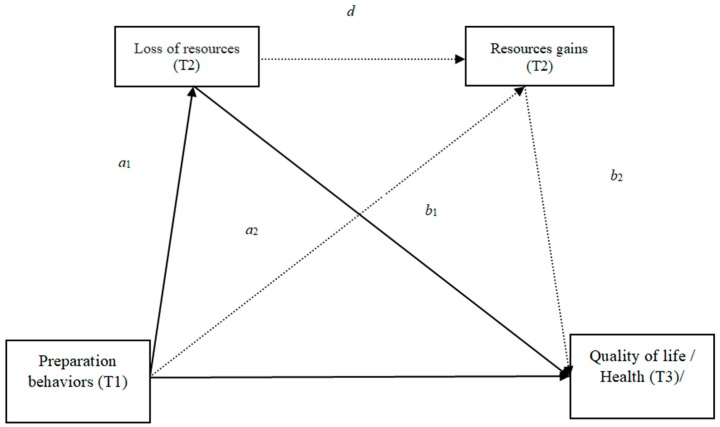
Visual representation of the study hypotheses. Hypothesis 1 (H1): Direct effect of the Behaviors of preparation in the Quality of Life (path c′); Hypothesis 2 (H2): Direct effect of Health Preparedness Behaviors (path c′); Hypothesis 3: Specific indirect effect through loss of resources in the Quality of life (path a1b1); Hypothesis 4: Specific indirect effect through the loss of resources in Health (path a1b1).

**Figure 2 ijerph-16-01539-f002:**
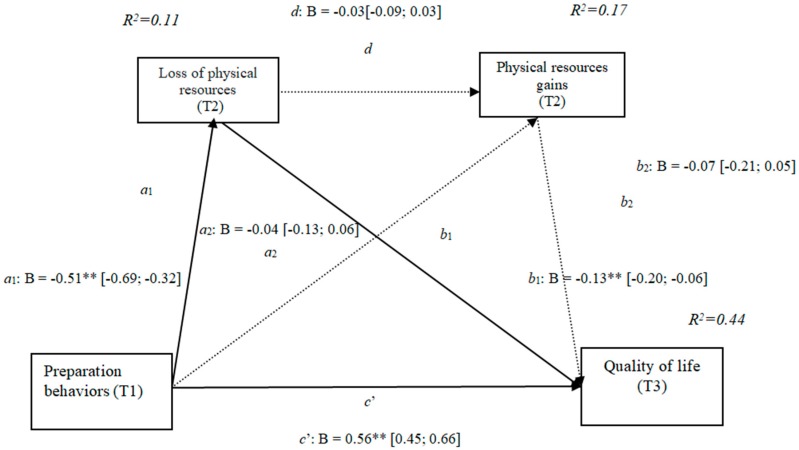
Non-standardized B coefficients, confidence intervals, and statistical significance for physical losses and gains. [95% CI]; * *p* < 0.05; ** *p* < 0.01.

**Figure 3 ijerph-16-01539-f003:**
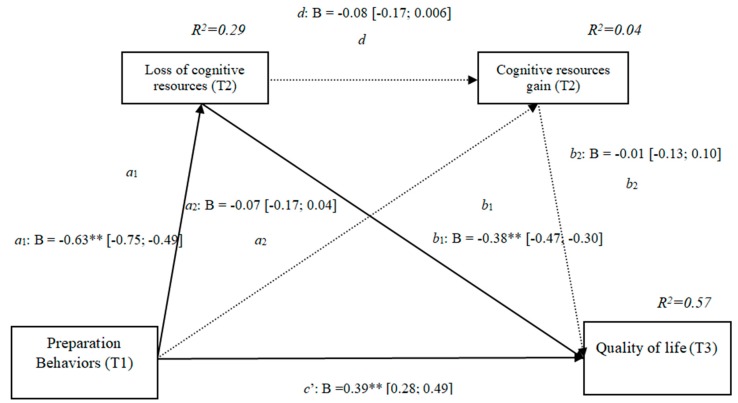
Non-standardized B coefficients, confidence intervals, and statistical significance for cognitive losses and gains. [95% CI]; * *p* < 0.05; ** *p* < 0.01.

**Figure 4 ijerph-16-01539-f004:**
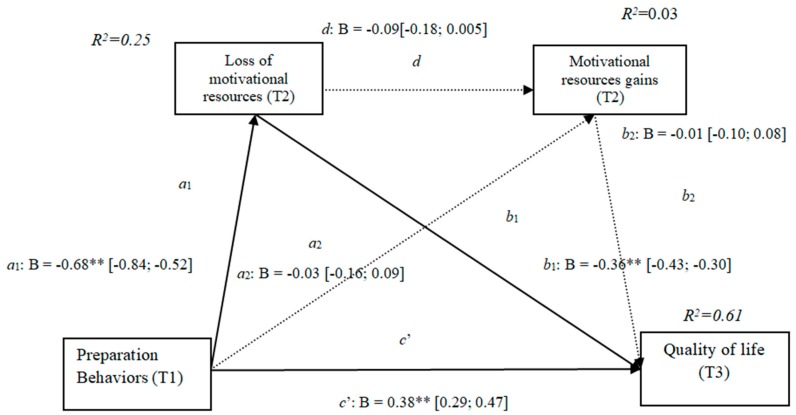
Non-standardized B coefficients, confidence intervals, and statistical significance for motivational losses and gains. [95% CI]; * *p* < 0.05; ** *p* < 0.01.

**Figure 5 ijerph-16-01539-f005:**
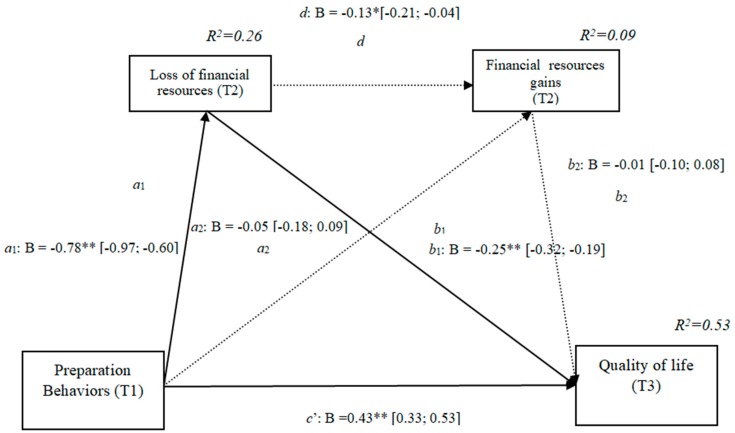
Non-standardized B coefficients, confidence intervals, and statistical significance for financial losses and gains. [95% CI]; * *p* < 0.05; ** *p* < 0.01.

**Figure 6 ijerph-16-01539-f006:**
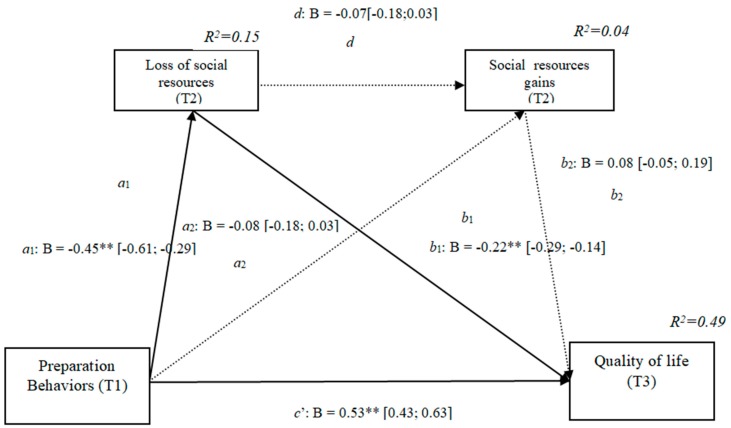
Non-standardized B coefficients, confidence intervals, and statistical significance for social gains and losses. [95% CI]; * *p* < 0.05; ** *p* < 0.01.

**Figure 7 ijerph-16-01539-f007:**
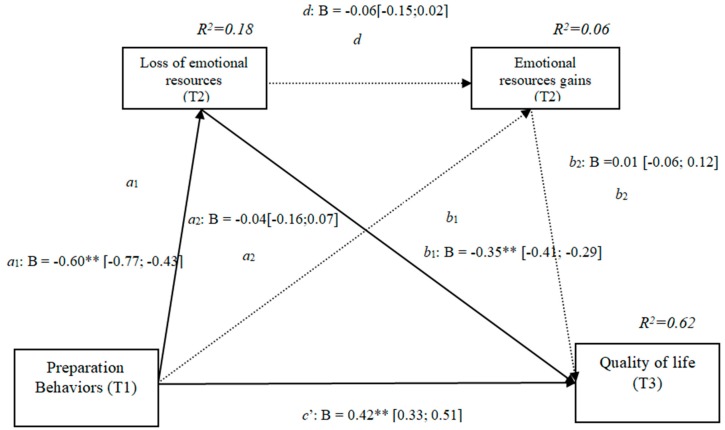
Non-standardized B coefficients, confidence intervals, and statistical significance for emotional losses and gains. Note: [95% CI]; * *p* < 0.05; ** *p* < 0.01.

**Figure 8 ijerph-16-01539-f008:**
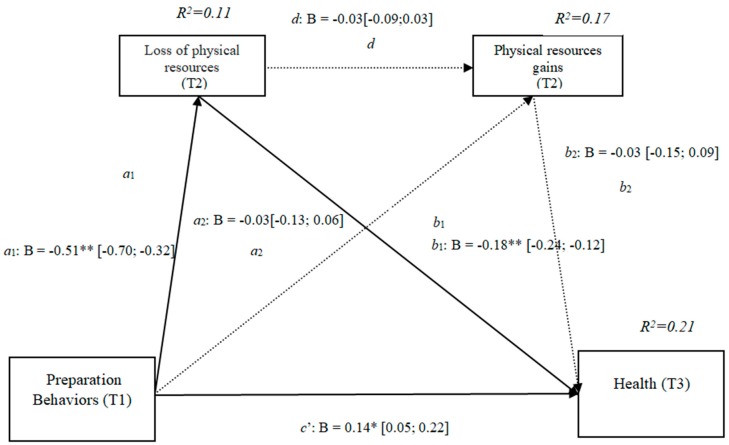
Non-standardized B coefficients, confidence intervals, and statistical significance for physical losses and gains. Note: [95% CI]; * *p* < 0.05; ** *p* < 0.01.

**Figure 9 ijerph-16-01539-f009:**
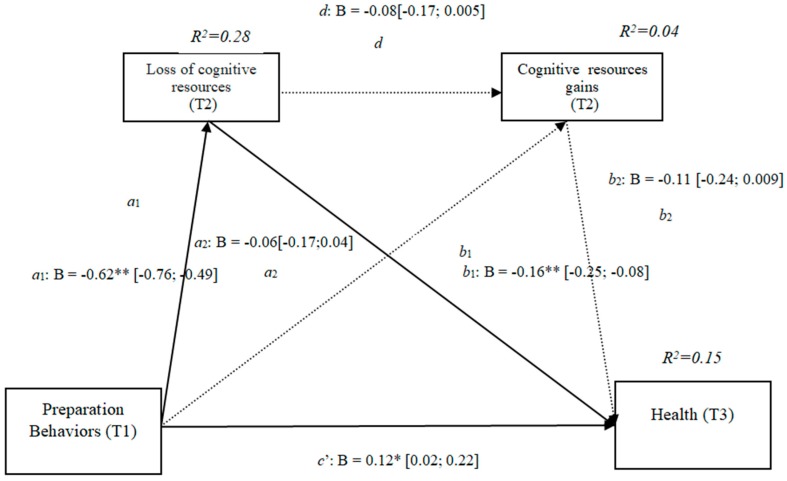
Non-standardized B coefficients, confidence intervals, and statistical significance for cognitive gains and losses. [95% CI]; * *p* < 0.05; ** *p* < 0.01.

**Figure 10 ijerph-16-01539-f010:**
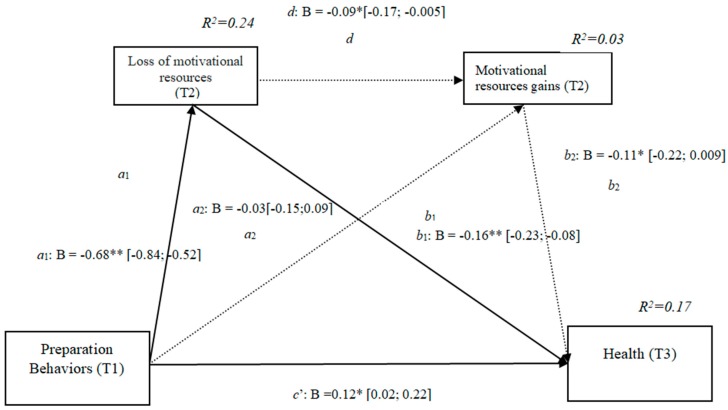
Non-standardized B coefficients, confidence intervals, and statistical significance for motivational losses and gains. [95% CI]; * *p* < 0.05; ** *p* < 0.01.

**Figure 11 ijerph-16-01539-f011:**
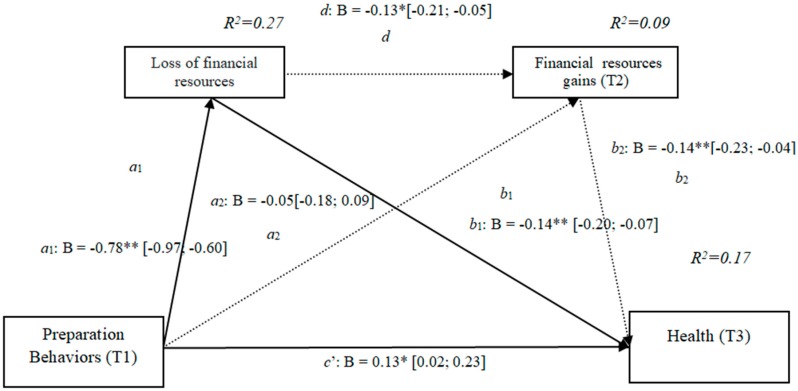
Non-standardized B coefficients, confidence intervals, and statistical significance for financial losses and gains. [95% CI]; * *p* < 0.05; ** *p* < 0.01.

**Figure 12 ijerph-16-01539-f012:**
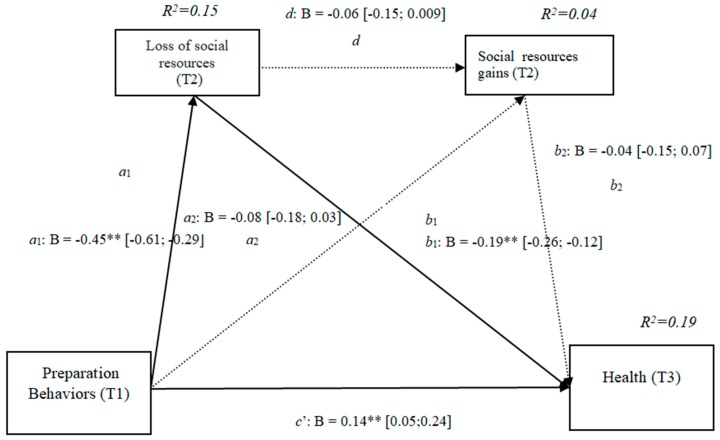
Non-standardized B coefficients, confidence intervals, and statistical significance for social losses and gains. [95% CI]; * *p* < 0.05; ** *p* < 0.01.

**Figure 13 ijerph-16-01539-f013:**
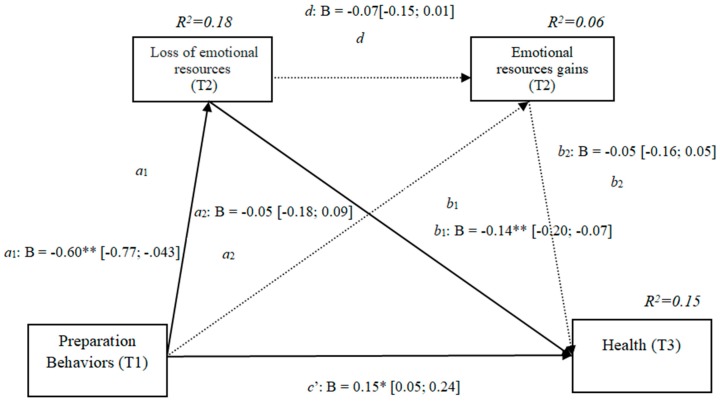
Non-standardized B coefficients, confidence intervals, and statistical significance for emotional losses and gains. [95% CI]; * *p* < 0.05; ** *p* < 0.01.

**Table 1 ijerph-16-01539-t001:** Descriptive statistics and correlation matrix of the variables of the study.

	M	SD	1	2	3	4	5	6	7	8	9	10	11	12	13	14	15	16	17	18
1. Age (T1)	62.08	4.88	1																	
2. Gender (T1)	(a)	(a)	0.01	1																
3. Seniority in employment(T1)	29.9	9.3	0.28 **	−0.02	1															
4. Number dependent(T1)	1.43	1.48	−0.09	0.00	−0.14 *	1														
5. Behaviors Retirement Preparation (T1)	3.33	0.65	0.04	0.01	0.05	−0.09	1													
6. Physical R. Losses(T2)	2.24	1.00	−0.00	0.05	0.01	0.01	−0.31 **	1												
7. Earnings R. Physical (T2)	3.47	0.49	−0.07	−0.38 **	−0.10	−0.01	−0.04	−0.07	1											
8. Losses R. Cognitive (T2)	2.10	0.80	−0.02	−0.06	0.07	0.14 *	−0.50 **	0.42 **	−0.02	1										
9. Earnings R. Cognitive(T2)	3.62	0.47	−0.04	0.01	−0.04	−0.12 *	−0.02	−0.02	0.55 **	−0.11	1									
10. Losses R. Motivational(T2)	2.21	0.93	−0.01	−0.07	0.08	0.08	−0.47 **	0.42 **	−0.00	0.80 **	−0.11	1								
11. Earnings R. Motivational (T2)	3.48	0.56	−0.05	0.01	−0.02	−0.09	0.04	−0.01	0.44 **	−0.13 *	0.85 **	−0.14 *	1							
12. Losses R. Financial (T2)	2.63	1.05	−0.01	0.01	−0.01	0.17 **	−0.48 **	0.30 **	−0.01	0.61 **	−0.06	0.62 **	−0.08	1						
13. Earnings R. Financial (T2)	3.09	0.63	0.09	−0.16 **	−0.01	−0.11	0.06	−0.02	0.45 **	−0.16 *	0.52 **	−0.12 *	0.55 **	−0.20 **	1					
14. Losses R. Social (T2)	2.02	0.87	−0.01	−0.02	0.11	0.14 *	−0.33 **	0.37 **	0.02	0.65 **	0.00	0.58 **	−0.04	0.45 **	−0.06	1				
15. Earnings R. Social (T2)	3.74	0.51	0.06	−0.06	−0.03	−0.10	−0.05	−0.07	0.53 **	−0.11	0.72 **	−0.08	0.63 **	−0.08	0.48 **	−0.11	1			
16. Losses R. Emotional (T2)	2.27	0.93	−0.05	−0.03	0.01	0.09	−0.42 **	0.42 **	−0.00	0.71 **	−0.11	0.80 **	−0.11	0.55 **	−0.20 **	0.50 **	−0.10	1		
17. Earnings R. Emotional (T2)	3.39	0.56	−0.17 **	−0.02	−0.06	−0.12	0.00	−0.04	0.56 **	−0.12	0.75 **	−0.11	0.74 **	−0.09	0.52 **	−0.03	0.62 **	−0.09	1	
18. Health (T3)	3.86	0.49	−0.01	−0.05	0.03	0.00	0.29 **	−0.42 **	0.00	−0.32 **	−0.09	−0.34 **	−0.08	−0.32 **	−0.10	−0.37 **	−0.01	−0.32 **	−0.03	1
19. Quality of life (T3)	3.51	0.65	0.01	0.08	−0.07	−0.10	0.62 **	−0.37 **	−0.07	−0.67 **	0.04	−0.70 **	0.08	−0.61 **	0.10	−0.48 **	0.06	−0.68 **	0.08	0.39 **

(**a**) This value does not have statistical significance because it is a dummy variable (0 = Man; 1 = Woman), * *p* < 0.05; ** *p* < 0.01.

**Table 2 ijerph-16-01539-t002:** Comparison of the indirect effects of preparation behaviors (T1) through physical resource losses (T2) and gains (T2) in the prediction of quality of life (T3).

Criterion	Variable: Quality of Life (T3)		*B*	*Boot*	*BootL*	*BootU*
Indirect Effects of Preparation Behaviors (T1)				*SE*	*LLCI*	*ULCI*
Total:			0.07	0.04	0.01	0.17
Ind1: Preparation behavior (T1) → lost resources physical (T2) → Quality of life (T3)			0.07	0.04	0.02	0.19
Ind2: Preparation behavior (T1) → lost resources physical (T2) → earnings resources physical (T2) → Quality of life (T3)			−0.00	0.00	−0.01	0.00
Ind3: Preparation behavior (T1) → earnings resources physical (T2) → Quality of life (T3)			0.00	0.01	0.00	0.02
(C1) Ind1	Less	Ind2	0.07	0.04	0.02	0.17
(C2) Ind1	Less	Ind3	0.06	0.04	0.02	0.16
(C3) Ind2	Less	Ind3	0	0.05	−0.03	0.00

T1 = Time 1; T2 = Time 2; T3 = Time 3; Ind = Indirect Effect. LLCI = lower level of the Confidence interval of 95%. ULCI = Upper level of the Confidence interval of 95%. SE = Standard Error. Indirect effects are significant when the interval does not contain the zero.

**Table 3 ijerph-16-01539-t003:** Comparison of the indirect effects of preparation behaviors (T1) through the losses (T2) and gains of cognitive resources (T2) in the prediction of quality of life (T3).

Criterion	Variable: Quality of Life (T3)		*B*	*Boot*	*BootL*	*BootU*
Indirect Effects of Preparation Behaviors (T1)				*SE*	*LLCI*	*ULCI*
Total:			0.23	0.04	0.15	0.33
Ind1: Preparation behavior (T1) →lost resources cognitive (T2) → Quality of life (T3)			0.23	0.04	0.15	0.33
Ind2: Preparation behavior (T1) →lost resources cognitive (T2) → earnings resources cognitive (T2) → Quality of life (T3)			−0.00	0.00	−0.01	0.00
Ind3: Preparation behabior (T1) → earnings resources cognitive (T2) → Quality of life (T3)			0.00	0.00	−0.01	0.01
(C1) Ind1	Less	Ind2	0.24	0.04	0.15	0.33
(C2) Ind1	Less	Ind3	0.23	0.04	0.16	0.33
(C3) Ind2	Less	Ind3	−0.00	0.01	−0.02	0.01

T1 = Time 1; T2 = Time 2; T3 = Time 3; Ind = Indirect Effect. Note. T1 = Time 1; T2 = Time 2; T3 = Time 3; Ind = Indirect Effect. LLCI = lower level of the Confidence interval of 95%. ULCI = Upper level of the Confidence interval of 95%. SE = Standard Error. Indirect effects are significant when the interval does not contain the zero.

**Table 4 ijerph-16-01539-t004:** Comparison of the indirect effects of preparation behaviors (T1) through motivational resource losses (T2) and gains (T2) in the prediction of quality of life (T3).

Criterion	Variable: Quality of life (T3)		*B*	*Boot*	*Boot*	*Boot*
Indirect Effects of Preparation Behaviors (T1)				*SE*	*LLCI*	*ULCI*
Total:			0.24	0.04	0.16	0.34
Ind1: Preparation behavior (T1) lost resources motivational (T2) → Quality of life (T3)			0.24	0.04	0.17	0.34
Ind2: Preparation behavior (T1) lost resources motivational (T2) → earnings resources motivational (T2) → Quality of life (T3)			−0.00	0.00	−0.01	0.00
Ind3: Preparation behavior (T1) earnings resources motivational (T2) → Quality of life (T3)			0.00	0.00	−0.00	0.01
(C1) Ind1	Less	Ind2	0.24	0.04	0.16	0.34
(C2) Ind1	Less	Ind3	0.24	0.04	0.16	0.34
(C3) Ind2	Less	Ind3	−0.00	0.01	−0.01	0.01

T1 = Time 1; T2 = Time 2; T3 = Time 3; Ind = Indirect Effect. LLCI = lower level of the Confidence interval of 95%. ULCI = Upper level of the Confidence interval of 95%. SE = Standard Error. Indirect effects are significant when the interval does not contain the zero.

**Table 5 ijerph-16-01539-t005:** Comparison of the indirect effects of preparation behaviors (T1) through financial resource losses (T2) and gains (T2) in the prediction of quality of life (T3).

Criterion	Variable: Quality of Life (T3)		*B*	*Boot*	*Boot*	*Boot*
Indirect Effects of Preparation Behaviors (T1)				*SE*	*LLCI*	*ULCI*
Total:			0.19	0.04	0.12	0.29
Ind1: Preparation behavior (T1) →lost resources financial (T2) → Quality of life (T3)			0.19	0.04	0.12	0.29
Ind2: Preparation behavior (T1) → lost resources financial (T2) → earnings resources financial (T2) → Quality of life (T3)			0.00	0.00	−0.01	0.01
Ind3: Conductas de preparación(T1) → earnings resources financial (T2) → Quality of life (T3)			−0.00	0.00	−0.01	0.01
(C1) Ind1	Less	Ind2	0.19	0.04	0.12	0.29
(C2) Ind1	Less	Ind3	0.19	0.04	0.12	0.29
(C3) Ind2	Less	Ind3	0.00	0.01	−0.01	0.01

T1 = Time 1; T2 = Time 2; T3 = Time 3; Ind = Indirect Effect. LLCI = lower level of the Confidence interval of 95%. ULCI = Upper level of the Confidence interval of 95%. SE = Standard Error. Indirect effects are significant when the interval does not contain the zero.

**Table 6 ijerph-16-01539-t006:** Comparison of the indirect effects of preparation behaviors (T1) through social resources losses (T2) and gains (T2) in the prediction of quality of life (T3).

Criterion	Variable: Quality of Life (T3)		*B*	*Boot*	*Boot*	*Boot*
Indirect Effects of Preparation Behaviors (T1)				*SE*	*LLCI*	*ULCI*
Total:			0.09	0.04	0.03	0.19
Ind1: Preparation behavior (T1) → lost resources social (T2) → Quality of life (T3)			0.09	0.03	0.04	0.19
Ind2: Preparation behavior (T1) → lost resources social (T2) → earnings resources social (T2) → Quality of life (T3)			0.00	0.00	−0.00	0.01
Ind3: Preparation behavior (T1) →earnings resources social (T2) →Quality of life (T3)			−0.01	0.01	−0.02	0.00
(C1) Ind1	Less	Ind2	0.09	0.03	0.04	0.19
(C2) Ind1	Less	Ind3	0.10	0.03	0.04	0.19
(C3) Ind2	Less	Ind3	0.01	0.01	−0.00	0.03

T1 = Time 1; T2 = Time 2; T3 = Time 3; Ind = Indirect Effect. LLCI = lower level of the Confidence interval of 95%. ULCI = Upper level of the Confidence interval of 95%. SE = Standard Error. Indirect effects are significant when the interval does not contain the zero.

**Table 7 ijerph-16-01539-t007:** Comparison of the indirect effects of preparation behaviors (T1) through losses (T2) and gains of emotional resources (T2) in the prediction of quality of life (T3).

Criterion	Variable: Quality of Life (T3)		*B*	*Boot*	*Boot*	*Boot*
Indirect Effects of Preparation Behaviors (T1)				*SE*	*LLCI*	*ULCI*
Total:			0.20	0.04	0.12	0.31
Ind1: Preparation behavior (T1) → lost resources emotional (T2) → Quality of life (T3)			0.20	0.04	0.12	0.31
Ind2: Preparation behavior (T1) → lost resources emotional (T2) → earnings resources emotional (T2) → Quality of life (T3)			0.00	0.00	−0.00	0.01
Ind3: Preparation behavior (T1) → earnings resources emotional (T2) → Quality of life (T3)			−0.00	0.00	−0.02	0.00
(C1) Ind1	Less	Ind2	0.20	0.04	0.12	0.31
(C2) Ind1	Less	Ind3	0.21	0.04	0.12	0.31
(C3) Ind2	Less	Ind3	0.00	0.01	−0.00	0.02

T1 = Time 1; T2 = Time 2; T3 = Time 3; Ind = Indirect Effect. LLCI = lower level of the Confidence interval of 95%. ULCI= Upper level of the Confidence interval of 95%. SE= Standard Error. Indirect effects are significant when the interval does not contain the zero.

**Table 8 ijerph-16-01539-t008:** Comparison of the indirect effects of preparation behaviors (T1) through physical resource losses (T2) and gains (T2) in the prediction of health (T3).

Criterion	Variable: Health (T3)		*B*	*Boot*	*Boot*	*Boot*
Indirect Effects of Preparation Behaviors (T1)				*SE*	*LLCI*	*ULCI*
Total:			0.09	0.04	0.04	0.17
Ind1: Preparation behavior (T1) → lost resources physical (T2) → Health (T3)			0.09	0.04	0.04	0.17
Ind2: Preparation behavior (T1) → lost resources physical (T2) → earnings resources physical (T2) → Health (T3)			−0.00	0.00	−0.01	0.00
Ind3: Preparation behavior (T1) → earnings resources physicists (T2) → Health (T3)			0.00	0.00	−0.00	0.01
(C1) Ind1	less	Ind2	0.09	0.04	0.04	0.18
(C2) Ind1	less	Ind3	0.09	0.04	0.04	0.17
(C3) Ind2	less	Ind3	−0.00	0.00	−0.02	0.00

T1 = Time 1; T2 = Time 2; T3 = Time 3; Ind = Indirect Effect. LLCI = lower level of the Confidence interval of 95%. ULCI = Upper level of the Confidence interval of 95%. SE = Standard Error. Indirect effects are significant when the interval does not contain the zero.

**Table 9 ijerph-16-01539-t009:** Comparison of the indirect effects of preparation behaviors (T1) through the losses (T2) and gains of cognitive resources (T2) in the prediction of health (T3).

Criterion	Variable: Health (T3)		*B*	*Boot*	*Boot*	*Boot*
Indirect Effects of Preparation Behaviors (T1)				*SE*	*LLCI*	*ULCI*
Total:			0.10	0.03	0.04	0.17
Ind1: Preparation behavior (T1) → lost resources cognitive (T2) → Health (T3)			0.10	0.03	0.04	0.17
Ind2: Preparation behavior (T1) → lost resources cognitive (T2) → earnings resources cognitive (T2) → Health (T3)			−0.01	0.01	−0.02	0.00
Ind3: Preparation behavior (T1) → earnings resources cognitive (T2) → Health (T3)			−0.01	0.01	−0.00	0.03
(C1) Ind1	less	Ind2	0.11	0.03	0.05	0.17
(C2) Ind1	less	Ind3	0.09	0.03	0.03	0.16
(C3) Ind2	less	Ind3	−0.01	0.01	−0.05	0.00

T1 = Time 1; T2 = Time 2; T3 = Time 3; Ind = Indirect Effect. LLCI = lower level of the Confidence interval of 95%. ULCI= Upper level of the Confidence interval of 95%. SE= Standard Error. Indirect effects are significant when the interval does not contain the zero.

**Table 10 ijerph-16-01539-t010:** Comparison of the indirect effects of preparation behaviors (T1) through losses (T2) and gains of motivational resource gains (T2) in the prediction of health (T3).

Criterion	Variable: Health (T3)		*B*	*Boot*	*Boot*	*Boot*
Indirect Effects of Preparation Behaviors (T1)				*SE*	*LLCI*	*ULCI*
Total:			0.10	0.026	0.06	0.16
Ind1: Preparation behavior (T1) → lost resources motivational (T2) → Health (T3)			0.11	0.026	0.06	0.16
Ind2: Preparation behavior (T1) → lost resources motivational (T2) → earnings resources motivational (T2) → Health (T3)			−0.01	0.01	−0.02	−0.00
Ind3: Preparation behavior (T1) → earnings resources motivational (T2) → Health (T3)			0.00	0.01	−0.01	0.02
(C1) Ind1	Less	Ind2	0.11	0.03	0.07	0.17
(C2) Ind1	Less	Ind3	0.10	0.03	0.05	0.16
(C3) Ind2	Less	Ind3	−0.01	0.01	−0.04	0.00

T1 = Time 1; T2 = Time 2; T3 = Time 3; Ind = Indirect Effect. LLCI = lower level of the Confidence interval of 95%. ULCI= Upper level of the Confidence interval of 95%. SE = Standard Error. Indirect effects are significant when the interval does not contain the zero.

**Table 11 ijerph-16-01539-t011:** Comparison of the indirect effects of preparation behaviors (T1) through financial resource losses (T2) and gains (T2) in the prediction of health (T3).

Criterion	Variable: Health (T3)		*B*	*Boot*	*Boot*	*Boot*
Indirect Effects of Preparation Behaviors (T1)				*SE*	*LLCI*	*ULCI*
Total:			0.10	0.03	0.04	0.16
Ind1: Preparation behavior (T1) → lost resources financial (T2) → Health (T3)			0.10	0.03	0.05	0.16
Ind2: Preparation behavior (T1) → lost resources Financial (T2) → earnings resources financial (T2) → Health (T3)			−0.01	0.01	−0.03	−0.00
Ind3: Preparation behavior (T1) → earnings resources financial (T2) → Health (T3)			0.00	0.01	−0.01	0.03
(C1) Ind1	Less	Ind2	0.12	0.03	0.06	0.18
(C2) Ind1	Less	Ind3	0.10	0.03	0.04	0.16
(C3) Ind2	Less	Ind3	−0.02	0.01	−0.06	−0.00

T1 = Time 1; T2 = Time 2; T3 = Time 3; Ind = Indirect Effect. LLCI = lower level of the Confidence interval of 95%. ULCI = Upper level of the Confidence interval of 95%. SE = Standard Error. Indirect effects are significant when the interval does not contain the zero.

**Table 12 ijerph-16-01539-t012:** Comparison of the indirect effects of preparation behaviors (T1) through the losses (T2) and gains of social resources (T2) in the prediction of health (T3).

Criterion	Variable: Health (T3)		*B*	*Boot*	*Boot*	*Boot*
Indirect Effects of Preparation Behaviors (T1)				*SE*	*LLCI*	*ULCI*
Total:			0.09	0.03	0.05	0.15
Ind1: Preparation behavior (T1) → lost resources social (T2) → Health (T3)			0.08	0.03	0.05	0.15
Ind2: Preparation behavior (T1) → lost resources social (T2) → earnings resources social (T2) → Health (T3)			−0.00	0.00	−0.01	0.00
Ind3: Preparation behavior (T1) → earnings resources social (T2) → Health (T3)			0.00	0.01	−0.00	0.02
(C1) Ind1	menos	Ind2	0.09	0.03	0.05	0.15
(C2) Ind1	menos	Ind3	0.08	0.02	0.04	0.14
(C3) Ind2	menos	Ind3	−0.00	0.01	−0.03	0.01

T1 = Time 1; T2 = Time 2; T3 = Time 3; Ind = Indirect Effect. LLCI = lower level of the Confidence interval of 95%. ULCI = Upper level of the Confidence interval of 95%. SE = Standard Error. Indirect effects are significant when the interval does not contain the zero.

**Table 13 ijerph-16-01539-t013:** Comparison of the indirect effects of preparation behaviors (T1) through the losses (T2) and gains of emotional resources (T2) in the prediction of health (T3).

Criterion	Variable: Health(T3)		*B*	*Boot*	*Boot*	*Boot*
Indirect Effects of Preparation Behaviors (T1)				*SE*	*LLCI*	*ULCI*
Total:			0.08	0.03	0.03	0.15
Ind1: Preparation behavior (T1) → lost resources emotional (T2) → Health (T3)			0.08	0.03	0.03	0.15
Ind2: Preparation behavior (T1) → lost resources emotional (T2) → earnings resources emotional (T2) → Health (T3)			−0.00	0.03	−0.01	0.00
Ind3: Preparation behavior (T1) → earnings resources emotional (T2) → Health (T3)			0.00	0.00	−0.00	0.02
(C1) Ind1	Less	Ind2	0.08	0.03	0.03	0.15
(C2) Ind1	Less	Ind3	0.08	0.03	0.03	0.15
(C3) Ind2	Less	Ind3	−0.00	0.01	−0.02	0.00

T1 = Time 1; T2 = Time 2; T3 = Time 3; Ind = Indirect Effect. LLCI = lower level of the Confidence interval of 95%. ULCI = Upper level of the Confidence interval of 95%. SE= Standard Error. Indirect effects are significant when the interval does not contain the zero.

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
