# Peer review of "Quality of Life and Health: Influence of Preparation for Retirement Behaviors through the Serial Mediation of Losses and Gains"

_ijerph, 2019, doi:10.3390/ijerph16091539_

Round 1
Reviewer 1 Report
This article is focused on a relevant and emerging field – “how the transition to retirement
significantly explain their well-being after retirement. “. There are promising sections in the paper but there are also opportunities of improvement. Let me address the most claiming of these weak sections:
Point 1: In section 1, the authors claim “because if people perceive that their resources have decreased in a vital facet, then they
will be forced to devote additional resources to compensate for the loss, even if such a loss is not
real.” Actually, authors must justify which losses are not real/may be not perceived as real. Who has suggested such assumption (e.g., “relevance”, “unrelatedness”, “theoretical focus”, etc) and which results the suggesting authors got? Additionally, authors must discuss better their major 2 Hypotheses. How can they separate Health from Quality of Life? And if they can, why do not they separate Quality of Life from other strictly related dimensions, like endowments, mobility, life satisfaction, emotional support, social connectedness, etc? Literature must be reinforced with other papers published about Iberian elderly populations, namelyMourao & Cilina (2018). ‘No country for old men’? The multiplier effects of pensions in Portuguese municipalities.December 2018. Journal of Pension Economics and Finance.DOI: 10.1017/S1474747218000318
Point 2: Tables and Figures are not well defined in the presented format. Authors must opt for a different format. There are also different types/styles of edition which is improper to a scientific submission.
In section 2., references about the goodness of the claim “H4: The relationship between retirement preparation behaviors (T1) and retirees' health (T3) will
be significantly mediated by the loss of physical, cognitive, motivational, financial, social, and
emotional resources, whereas gains in physical, cognitive, motivational, financial, social, and
emotional resources will not be significant mediators “are missing as well references regarding other unclear statements along the Methodological sections.
Point 3. The paper then flows for a kind of descriptive analysis combined with trials of Confirmatory Factor Analysis (CFA) in which previously identified dimensions cover some of the found evidences. I will appreciate to have the traditional measures connected to CFA also exhibited and commented. Finally, I will also appreciate to have more information about the questionnaire, its distribution for the respondents, the discussion of pre-tests and along-checks, as well as an extended discussion of similar structures with their own weaknesses.
Point 4. Conclusions are anecdotal (in length and in content).
Author Response
Dear reviewer, thanks for your helping suggestions and comments. We have tried to improve our paper accordingly.
We have prepared a document explaining our changes.

Reviewer 2 Report
- There is a fairly large literature in economics that considers well-being and health in the transition to retirement. A couple of relevant papers include the following, and the papers cited within them:
Fitzpatrick, Maria D., and Timothy J. Moore. 2018. “The Mortality Effects of Retirement: Evidence from Social Security Eligibility at Age 62.” Journal of Public Economics 157: 121-137.
Maestas, Nicole. 2010. “Back to Work: Expectation and Realization of Work after Retirement.” Journal of Human Resources 45(3): 718-748.
- The drop-off in response rates between the initial survey and the third follow-up may be somewhat concerning. The authors should analyze whether attrition subsequent to the first round of the survey is correlated with observable characteristics of the respondents. I should like to see no significant correlation of future attrition with measurable characteristics. If significant correlations do exist the authors should discuss how this might bias their estimates.
- The theoretical discussion in Section 1 is dense, and can be streamlined.
- The table notes should describe the analysis more completely, and the table headings should also be defined where not obvious (e.g., lower and upper bounds for the confidence intervals).
- It was not clear to me where the descriptions of results in the text come from. For example, in section 3.2 the authors write that the indirect effect of physical losses on quality of life was in the range of [0.02,0.17], however in Table 2 the CI appears to be [0.02,0.19]. Either I am not looking in the right place, or the text is incorrect, however a similar pattern is repeated essentially in the entire results section so I suspect I am misunderstanding something about the presentation of results in the tables. The authors should more clearly direct readers where to look in the tables or the figures for the results described in the text (e.g., “columns 3 and 4 of Table 3”, or “Figure 2”).
- In section 3.9 the authors describe the CI for the indirect effect of cognitive resource losses, and say it does not include 0, however then they write that the interval is [1.25,-0.07]. This seems to be a typographical error; I suspect the authors meant [-0.25,-0.07].
Author Response
Dear reviewer, we are very grateful for your comments and suggestions.
We have tried to improve our manuscript accordingly.
We have prepared a table with our changes and explanations.

Round 2
Reviewer 1 Report
The changes have been acceptable.